# Chronobiology of Cancers in the Liver and Gut

**DOI:** 10.3390/cancers16172925

**Published:** 2024-08-23

**Authors:** Jessica M. Ferrell

**Affiliations:** Department of Integrative Medical Sciences, Northeast Ohio Medical University, Rootstown, OH 44272, USA; jfrancl@neomed.edu; Tel.: +1-330-325-6468

**Keywords:** circadian rhythms, chronotherapy, hepatocellular carcinoma, cholangiocarcinoma, colorectal cancer

## Abstract

**Simple Summary:**

The circadian clock controls the rhythmic timing of nearly every aspect of physiology, including sleep, body temperature, hormone fluctuation, and overall metabolism on a daily basis. Research shows that disruption to this control contributes to the development of diseases, including cancer. Evidence also suggests that cancers may be amenable to treatments that target the circadian clock, which maximizes benefits while minimizing side effects. Here, the rhythmic influence over cancers of the liver and gut will be reviewed, and treatments that take into account the circadian clock will be discussed.

**Abstract:**

Circadian rhythms dictate the timing of cellular and organismal physiology to maintain homeostasis. Within the liver and gut, circadian rhythms influence lipid and glucose homeostasis, xenobiotic metabolism, and nutrient absorption. Disruption of this orchestrated timing is known to negatively impact human health and contribute to disease progression, including carcinogenesis. Dysfunctional core clock timing has been identified in malignant growths and may be used as a molecular signature of disease progression. Likewise, the circadian clock and its downstream effectors also represent potential for novel therapeutic targets. Here, the role of circadian rhythms in the pathogenesis of cancers of the liver and gut will be reviewed, and chronotherapy and chronopharmacology will be explored as potential treatment options.

## 1. Introduction

Circadian rhythms refer to physiological processes that occur over a period of approximately 24 h. They serve to synchronize cellular physiology with the external environment and ensure coordinated physiology within and across tissues in the body. These rhythms are generated and maintained by the endogenous biological clock, the suprachiasmatic nucleus (SCN) in the mammalian hypothalamus, and are influenced by stimuli called Zeitgebers (German: time-giver). The most important Zeitgeber is environmental light, though other non-photic Zeitgebers like physical activity, social interactions, and food are all capable of influencing circadian timing [1,2]. The endogenous circadian period in humans is approximately 24 h 11 min; thus, morning environmental light is critical for entraining, or synchronizing, the clock to prevent daily drift in timing. Photic stimuli are received by the SCN from intrinsically photosensitive retinal ganglion cells via the retinohypothalamic tract. This input influences the timing of neuronal firing and the expression of the core clock genes that make up the transcriptional-translation feedback loop (TTFL) within the SCN. SCN timing is then propagated through neurohormonal connections to other brain regions and ultimately throughout the body.

Circadian rhythms exist in nearly every living organism, with homologous genes and proteins comprising the TTFL (Figure 1). In mammals, the clock proteins circadian locomotor output cycles kaput (CLOCK) and brain-and-muscle ARNT-like 1 (BMAL1) heterodimerize to induce transcription of the additional clock proteins Period 1 and 2 (PER 1/2) and Cryptochrome 1 and 2 (CRY 1/2). PER and CRY translocate to the nucleus and heterodimerize to inhibit the CLOCK/BMAL1 dimer. This inhibits *PER*/*CRY* gene transcription and, thus, forms a negative feedback loop. PER and CRY are subject to phosphorylation by casein kinase 1 (CK1), leading to their ubiquitin-mediated degradation and completing the 24 h cycle. Additional regulation by the nuclear receptors retinoid-related orphan receptor a (RORα) and REV-ERBα serves to induce or suppress *BMAL1* transcription, respectively [3]. The TTFL in the SCN transmits this rhythm via the expression of clock-controlled genes and through the release of neuropeptides and neurotransmitters that synchronize the TTFLs in other tissues to brain (and ultimately, environmental) timing. This ensures an organism exhibits appropriate anticipation and physiological adaptation to metabolic changes throughout the day.

Cellular metabolic demands change with nutrient intake, sleep–wake cycles, and hormonal fluctuations, and properly synchronized rhythms ensure these needs are met. The liver and gut play a front-line role in nutrient regulation due to their functions in first-pass metabolism and nutrient absorption, respectively. As such, these organs are highly regulated by central and local circadian influence [4,5]. Desynchronization of central and peripheral timing can occur with disruption to sleep–wake cycles (i.e., chronic sleep loss, shift work, or jet lag) or poor diet and nutrient intake (i.e., fat, sugar, or alcohol), which contributes to the pathogenesis of metabolic syndrome and cardiovascular disease, neurodegenerative disease, and cancer [6]. Here, cancers of the liver and gut will be reviewed under a circadian lens, and the potential for time-based treatment of cancers (chronotherapy and chronopharmacology) will be explored.

## 2. Circadian Rhythms in the Liver and Gut

The liver plays a central role in maintaining normoglycemia and lipid homeostasis during feeding and fasting, and in processing xenobiotics for distribution and excretion. Circadian rhythms directly or indirectly influence these processes, and studies in animals confirm that approximately 10–25% of the liver transcriptome and proteome are regulated by circadian genes [7,8,9]. This translates to a cellular environment that is widely governed by rhythmic input.

### Consequences of Rhythm Disruption

Disruption of circadian rhythms remains a significant physiological risk factor for the development of breast, prostate, gastrointestinal and colon cancers, among others [10,11,12,13]. The pathological mechanisms are not clear, though temporal dysregulation of the immune system, endocrine system, and DNA repair systems are likely involved. Animal models with clock disturbances, whether due to SCN ablation, genetic knockout of clock genes, or exposure to circadian disruption, provide insight into the reciprocal relationships between clocks and cancer progression.

Shift workers represent a subset of the population exposed to chronic circadian disruption. Forced desynchronization of sleep–wake cycles from the external environment alters the cellular timing of nutrient availability and use, hormone production (including melatonin, which drives sleep), and response to external stimuli that subsequently manifests as metabolic pathology. The type and duration of shift work may also affect oncogenesis. Several studies have demonstrated that the risk of breast cancer is significantly increased in women who engage in long-term shift work [14,15]. Interestingly, one recent meta-analysis that included more than 1 million subjects found a significant association only between short-term night shift work and the development of breast cancer [16] and another smaller study corroborated these findings in individuals engaged in shift work for less than 5 years [17]. Similar associations were found between long-term shift work and prostate cancer [13,18], colorectal cancer [19], and digestive cancers [20].

However, some analyses refute these claims [21,22]. Inconsistency in defining work hours, lack of control over sleep quality and duration, and the potential for healthy worker bias may all contribute to differences in study results. Still, combined evidence in animal and human studies led the International Agency for Research on Cancer to designate shift work as a Class 2A probable human carcinogen, and this designation was maintained in a recent review of the evidence in 2019 [23].

Position-in-time-zone studies offer another real-world model of chronic circadian disruption. Geographical time zones were originally centered on longitudinal meridians of the globe, such that the sun is overhead at approximately noon and the eastern and western edges of each zone are no more than 30 min misaligned from solar noon. However, modern manipulation of time zones in the United States extended these boundaries such that people, particularly those living in Western edges of time zones, are misaligned by 60 min or more (i.e., the sun is overhead at 1 pm or later). This misalignment of solar vs. social time represents a model of chronic circadian disruption and can be used to glean information about the detrimental effects of rhythm disruption. A review of over 4 million cancer diagnoses in the United States determined that the risk for developing leukemia and other cancers including stomach, liver, prostate, colorectal, lung, and breast cancer significantly increased from east-to-west within a time zone, even after statistical correction for the state, poverty level, and smoking [24]. Another study demonstrated a significantly increased risk of developing hepatocellular carcinoma (HCC) for every 5° of westward location within a time zone [25]. However, one recent study disputing these results found that only the incidence of prostate cancer increased from east to west within time zones [26].

## 3. Circadian Rhythms and Cancers of the Liver and Gut

### 3.1. Hepatocellular Carcinoma

Hepatocellular carcinoma (HCC) is the most common type of liver cancer and arises most often from chronic viral hepatitis infection or alcohol-associated liver disease [27]. Alterations in core clock genes in humans may contribute to susceptibility to HCC or resistance to its treatment. A recent small study of nearly 600 cases of HCC found increased expression of casein kinase 1d (*CSNK1D*) or reduced expression of *PER2* or *REV-ERBA* in tumor samples isolated from HCC patients [28]. Another study determined that the reduced expression of *PER1*, along with changes in several other clock-controlled genes, could predict the overall survival of patients with HCC [29]. This was confirmed in other studies that identified reduced expression of *PER1/2*, *CRY2*, or *RORA* in tumors of HCC patients that then correlated with poor survival [30,31]. Additionally, some single-nucleotide polymorphisms (SNPs) in *PER1/3* and *CRY1* were associated with both HCC survival and HCC recurrence, with the wild-type allele of *PER3* being the primary risk factor in predicting HCC recurrence [32]. A recent systematic review covering 25 studies also determined that patients carrying a variant allele of *PER3* had a significantly reduced risk of death and increased survival time compared to patients carrying homozygous wild-type alleles [33]. The knockout of clock genes in animal models provides some insight into the pathogenic mechanisms of HCC. Mice lacking *Per1/2*, *Cry1/2,* or *Bmal1* exhibited an increased occurrence of HCC that was preceded by steatosis and that was worsened by chronic jetlag. It was also determined that chronic jetlag contributed to the progression of metabolic derangement from steatosis to inflammation/fibrosis in mice in the absence of high-fat-diet administration or gene mutation, indicating that circadian disruption is an independent risk factor that triggers tumor initiation [34].

Core clock genes may be directly involved in hepatic cell proliferation. In vitro studies conducted in human hepatoma lines determined that *Bmal1* and *Clock* expression are required for proliferation and that silencing either gene induced apoptosis and cell cycle arrest. The BMAL1:CLOCK protein complex also suppressed cell cycle regulators and promoted the expression of oncogenes [35]. However, in a model of regeneration by partial hepatectomy, hepatocyte-specific knockout of *Bmal1* in mice inhibited mitotic signaling and cell growth [36]. Mice exposed to a chemical model of HCC exhibited significant reductions in hepatic *Per2* and *Cry1* mRNA concurrently with a significant increase in the number of proliferating liver cells [37]. In vitro overexpression of *CSNK1D* in HepG2 cells enhanced cell migration and proliferation, while silencing reduced proliferation and treatment resistance.

### 3.2. Cholangiocarcinoma and Cholangitis

Cholangiocarcinoma (CCA) is a rare malignant cancer arising from the biliary tree in the liver. The role of circadian rhythms in the progression of CCA is less clear, though some evidence points to clock involvement. A recent Mendelian randomization analysis of published genome-wide association studies found a negative correlation between biliary cancer and sleep duration and a positive correlation between daytime napping or insomnia and biliary cancer [38]. Expression of *PER1* was significantly reduced in human CCA cell lines and in biopsied samples from patients with CCA. The expression of other core clock genes in CCA cell lines was variable, while *BMAL1* was increased in CCA tissue samples and *CRY/CLOCK* was unchanged [39]. Importantly, this study demonstrated that rescue of *PER1* expression reduced cell proliferation and induced apoptosis. Another study utilizing the chemical induction of HCC in *Cry1/Cry2* double-knockout mice found a nearly 8-fold increase in the number of CCA compared to HCC, with no difference in the incidence of cancer type in wild-type mice [40]. This drastic shift from HCC to CCA in mutant mice was attributed to the role of Cry1/2 in the progression of CCA over the initiation of carcinogenesis.

Primary biliary cholangitis (PBC) is a biliary tract disease that presents with cholestasis and is associated with an increased risk of HCC [41]. Patients with PBC experience sleep fragmentation and reduced sleep quality, in part due to well-documented pruritus that increases at night [42]. A small study was conducted in patients with PBC exposed to bright light therapy. Just 15 days of therapy improved sleep quality and sleep timing, and rhythms of melatonin metabolites were more robust [43]. In relation to this, pinealectomy or chronic light exposure (both of which reduce circulating melatonin) following bile duct ligation to simulate PBC in rats worsened ductular reaction, fibrosis, and angiogenesis [44]. Conversely, rats receiving bile duct ligation followed by housing in total darkness had increased circulating melatonin with reduced fibrosis and biliary proliferation [45]. Together, these data point to strong roles for Period and Cryptochrome, and potentially their regulation by Casein Kinase, in the development of HCC and CCA.

### 3.3. Colorectal Cancer

Cancer affecting the large intestine (colorectal cancer/CRC) is the second-leading cause of death due to malignant tumors, and its prevalence is increasing in younger adults in the United States [46] and worldwide [47]. Within the gut, circadian rhythms control fluid and nutrient absorption, motility, and enzyme secretion [48]. Shift workers have an increased risk of developing colorectal cancer [10,19,49] and this pathology may involve direct shifting of the physiological rhythms in the colon or indirect effects on the microbiome. Chronic circadian disruption that mimicked shift work in mice resulted in increased colonic permeability and reduced expression of tight junction proteins. Shifted mice also had increased variability in the period and phase of Per2 rhythms in colon tissue, indicating that circadian disruption can directly affect local rhythms in the gut [50]. Inducing sleep fragmentation in mice accelerated chemically induced colon carcinogenesis, resulting in increased reactive oxygen species, oxidative DNA damage, and tumor size compared to control mice [51]. The core clock gene *Bmal1* may play a role in mediating CRC progression. Loss of *Bmal1* in a mouse model of human CRC accelerated tumor development and increased intestinal proliferation, possibly via increased Yap activity. However, this study also noted that *Bmal1* deletion alone was not sufficient to induce CRC in mice [52]. In vitro studies in several CRC cell lines also indicate that BMAL1 may play a role in carcinogenesis. *BMAL1* knockdown in two varieties of stable CRC cells (HCT116 and SW480) induced mTOR and p53, proliferation, and senescence, while its knockdown in a metastatic CRC line (SW620) induced proliferation in a p53-independent manner [53]. Another study examined *BMAL1* knockdown and determined that these cell lines express differentially altered levels of BMAL1 and CLOCK mRNA and protein and that BMAL1 promotes cell migration through the stimulation of exosome release [54]. BMAL1 expression was found to be reduced in CRC tumors compared to normal tissue, corroborating in vitro data [55]. However, at least one study demonstrated that *BMAL1* knockdown reduced cell migration and increased expression of cell adhesion molecules in HCT116, SW480, and SW620 cells [56]. Finally, in an analysis of more than 600 CRC samples, it was found that *PER1/3*, *CRY2,* and *BMAL1* were significantly suppressed and that increased *CSNK1D* was linked to shorter disease-free survival [57].

The gut microbiota encompasses the microorganisms that reside in the gastrointestinal tract. A recent comprehensive study examined the mechanistic interactions between CRC, the microbiome, and disrupted rhythms in mice. Fecal samples from chronically jet-lagged mice exhibited a loss in rhythmic microbiota flora, and levels of Clostridiales and Lachnospiraceae were reduced while Muribaculaceae levels were doubled. The transplantation of fecal samples from these mice to control mice aggravated CRC metastasis, and this was recapitulated in *Bmal1* knockout mice and *Per1/2* double-knockout mice [58]. Interestingly, this study also demonstrated that patients with higher *BMAL1* expression in tumor-infiltrating lymphocytes had a better survival prognosis than patients with lower *BMAL1* expression. Food intake, a nonphotic stimulus, can shift peripheral rhythms, and altered timing of food intake can disrupt liver and gut metabolism [59]. This was demonstrated in mice fed a regular chow diet in the light (rest period) or dark (active period) for one week; colon rhythms were shifted while central SCN rhythms were unchanged. When further treated with alcohol, CRC-model mice fed at the wrong time had increased polyps in number and size, which correlated with intestinal dysbiosis and increased inflammation that preceded polyposis [60]. Independent of food intake, patients with rectal cancer and self-reported sleep disturbances were found to have enriched *Dialister* and *Fusobacterium*, with reduced *Turicibacter* compared to rectal cancer patients without sleep disturbances [61].

## 4. Chronotherapy for Cancers of the Liver and Gut

Chronotherapy can refer to either the manipulation of sleep–wake rhythms in patients to improve health or the application of pharmaceutical treatments at optimized times. The circadian clock influences many aspects of drug absorption, distribution, metabolism, and excretion (Figure 2). Physiologically, gastric emptying, motility, and blood flow are partly regulated by circadian rhythms [62]. Intestinal efflux transporters protectively expel drugs into the lumen and represent one of the first barriers to drug bioavailability. These exporters are regulated rhythmically, and intestinal absorption of their substrates is reduced at the time of day in which exporter expression is highest [63,64]. Circadian rhythms change blood flow, with increased cardiac output and renal flow in the morning and daytime that decreases at night [65]. Hepatic metabolism via cytochrome P450 enzymes has been shown to be regulated by Clock [66], Bmal [67], Period [68,69], and other clock components. Finally, renal elimination of drugs, via the regulation of blood flow, tubular secretions, and urinary pH, also contributes to rhythmic variations in the pharmacodynamics of drugs and may be more efficient during the active phase [70].

Chronotherapy has been applied toward the treatment of hypertension and hyperlipidemia [71], major depressive disorder and seasonal affective disorder [72,73], obesity and Type 2 diabetes [74,75], and several cancers [76,77]. Adjusting the timing of therapeutic delivery to match the specific drug target allows for enhanced outcomes and often reduced off-target effects and detrimental side effects. One of the earliest implementations of chronotherapy was for the treatment of hypercholesterolemia with simvastatin, an inhibitor of the cholesterol synthesis enzyme Hmg-CoA reductase. It was discovered that patients given simvastatin in the evening exhibited significantly greater reductions in serum cholesterol compared to patients given the same doses in the morning [78,79]. This enhanced response to nighttime simvastatin is likely due to the circadian rhythm in Hmg-CoA reductase, which exhibits a well-documented peak in expression and activity at night [80]. Another seminal example is the use of chemotherapeutics in patients with metastatic CRC. Patients who received timed infusions of the chemotherapy drugs fluorouracil (early morning) and oxaliplatin (afternoon), coupled with folinic acid, showed a 76% increase in response rate with 4-fold fewer toxic side effects and fewer patient withdrawals from the study compared to patients receiving the same drugs in a constant infusion. Consequently, timing the administration of these drugs to increase tolerability also allowed for the dose of fluorouracil to be increased by 40%, which likely enhanced its efficacy [81]. These studies and others have paved the way for the application of chronotherapy for several other disorders, including other cancers.

### 4.1. Chronotherapy in the Liver

HCC often requires multidisciplinary care and surgical resection is the most common and preferred treatment, followed by systemic therapies reserved for patients with non-resectable HCC [82]. Current research is focused on finding circadian gene signatures for liver cancers including HCC and determining how to use those signatures to maximally target tumor growth. A recent analysis of published metadata confirmed that *PER1* expression was suppressed in HCC, while *CLOCK* and *RORA* were induced. These authors then developed a novel gene signature, comprised of *PER1, RORA, REV-ERBA*, and the additional clock genes *NPAS2* and *TIMELESS*, which independently predicted prognosis in patients with HCC [83]. Another analysis of single-cell transcriptome datasets from liver cancer samples determined that disruption in circadian gene expression was highly correlated to malignancy and reduced immune infiltration with variable immune responses [84]. These gene signatures can then be correlated to actionable targets of FDA-approved drugs, as was performed in a recent study that identified clock gene associations with sensitivity to fluorouracil (*CRY2*, *PER1/2*, and others) vs. resistance to fluorouracil (*NPAS2*, *CLOCK*, and others) [85].

Melatonin is produced in the pineal gland and increases at night to drive the timing of sleep. It is also a powerful antioxidant that inhibits inflammation and improves liver function. A small study of 32 patients found that circulating melatonin was reduced in patients with HCC or prostate cancer, and this could be correlated to poorer survival in both groups compared to healthy controls. Interestingly, these patients also self-reported worsened sleep quality compared to controls [86]. Another study similarly determined that the survival of patients with HCC after liver transplant also correlated with having higher melatonin prior to transplantation [87]. Melatonin was also shown to induce apoptosis and decrease viability in a human CCA cell line [88], and it decreased tumor volume and prevented mitochondrial damage in a rodent model of chemically induced CCA [89].

The effectiveness of radiation therapy may also be improved by timed delivery. In mice with chemically induced HCC, radiation delivered late in the active phase maximally inhibited proliferation in tumor cells but had low antiproliferative effects in non-tumor cells. This study also showed that radiation at this time did not affect clock gene expression in non-tumor cells and caused minimal effect on DNA double-strand breaks in these cells [90].

Time-restricted eating (TRE) refers to limiting ad libitum food consumption to specific hours of the day, usually during the active period. TRE has been studied for the management of obesity, Type 2 diabetes, and other symptoms of metabolic syndrome [59], though the exact mechanism by which TRE benefits metabolism is not entirely known. TRE often naturally entails caloric restriction, thus delineating the contribution of limiting time vs. limiting calories is also unclear. TRE has been utilized to mitigate cancer, as evidenced by several studies demonstrating its benefits in treating HCC. A genetic model of mice that developed spontaneous HCC when fed a TRE-high-fat diet had fewer liver tumors with significantly reduced volume compared to free-fed mice given the same diet [91]. Another study on rats with chemically induced HCC found that tumor tissue was less sensitive to the phase-shifting effects of TRE, though these results may be confounded by the combination of caloric restriction and TRE used in this study [92]. Timed caloric restriction in a rat model of HCC reduced hepatomegaly, cirrhosis progression, and potential for metastasis, but importantly, did not affect overall tumor count. This method also significantly induced *Bmal1* expression in the liver, which may act as a regulator of tumor suppression [93].

First-line systemic therapy for non-resectable HCC is atezolizumab plus bevacizumab. Interestingly, both agents may be subject to circadian influence. A recent study evaluated patients with HCC receiving atezolizumab +/− bevacizumab either in the morning or afternoon and found that median overall survival was significantly lower in the afternoon group [94]. Adjusting the timing of this therapy represents an additional clinical approach to improving outcomes, as the overall objective response rate to this combination was determined to be approximately 38% without chronomodulation [95]. Interestingly, high *BMAL1* expression correlated with a reduced clinical response to bevacizumab, including reduced progression-free survival and overall survival. It was also determined that *BMAL1* upregulated vascular endothelial growth factor A, the molecular target of bevacizumab, which may contribute to resistance [96]. Likewise, a study that examined the pharmacokinetics of immune checkpoint inhibitors, including atezolizumab, found that morning infusions were associated with increased overall survival and better response compared to afternoon delivery [77]. Given the relatively long half-life of atezolizumab (~27 d) and bevacizumab (~20 d), the exploration of novel delivery systems like chronomodulated infusion pumps may further enhance the therapeutic effects of these drugs.

### 4.2. Chronotherapy in the Gut

As mentioned, chronotherapy has been studied for the treatment of CRC. A trial investigating the effectiveness of chronotherapeutic fluorouracil, oxaliplatin, leuvocorin, and irinotecan (FOLFOXIRI) found that the timed administration of these drugs resulted in overall survival of nearly 20 months with nearly 9 months of progression-free survival [97]. Another study investigated the sex-specific differences in sensitivity to irinotecan toxicity and found that delivery time should be early morning for males but afternoon for females [98]. Mathematical models have been employed to further predict differences in efficacy and safety based on a patient’s individual rhythms, though it is noted that these models are based on CRC cell lines and mouse tissue [99,100]. Fasting suppresses glycolysis and reduces CRC proliferation, and may also shift the gut microbiome to increase the growth of beneficial bacteria like *Lactobacillus*, though the timing of caloric restriction and effects on peripheral clocks were not evaluated [101,102]. Interestingly, a study of patients with CRC found that those with a longer eating window or higher meal frequency exhibited less fatigue and had overall better outcomes [103]. A new trial in young adults with early-onset CRC (NCT06022887) will determine if an 8 h daytime eating window affects colonic inflammation or the gut microbiome; this trial was actively recruiting at the time of publication [104]. These sparse results highlight both the advantages of chronomodulation and the need for further investigation.

### 4.3. Clinical Applications

The application of chronotherapeutics is a budding science [105]. One of the simplest examples of the implementation of chronotherapy is the use of light devices to re-entrain the central clock. Light therapy involves the application of bright light targeted to specific times of the circadian day and was shown to improve seasonal and non-seasonal depression [106,107], cognition, and sleep quality in patients with early-stage Alzheimer’s disease [108], as well as insomnia [109]. Bright-light therapy also reduces fatigue in patients with cancer [110]. The application of chronotherapy for cancers of the liver and gut presents significant challenges and can involve re-entraining a dysfunctional clock in local tissues and/or optimizing the timing of drug delivery. Personalized monitoring of clocks in cancer cells via biopsy or molecular modeling may greatly improve disease-free and survival outcomes while simultaneously reducing side effects and improving quality of life, but this also represents a timely and economical burden. However, given that the cost of treatment for HCC or CRC in the United States was recently estimated to be $80,000–$290,000 per patient [111,112,113], investing in pre-clinical and translational research to precisely target circadian clocks in cancer is a logical response to these growing burdens.

## 5. Conclusions

Circadian rhythms influence nearly every biological process in humans. Despite their ubiquity, they are often overlooked in disease research. Understanding how biological rhythms function to drive molecular, cellular, and tissue physiology is crucial to developing targeted therapeutic approaches, novel biomarkers, and predictive models to better treat diseases, including cancers of the liver and gut. Chronotherapy is an understudied aspect of circadian biology that represents an exciting research opportunity for precision medicine in oncology and beyond.

## Figures and Tables

**Figure 1 cancers-16-02925-f001:**
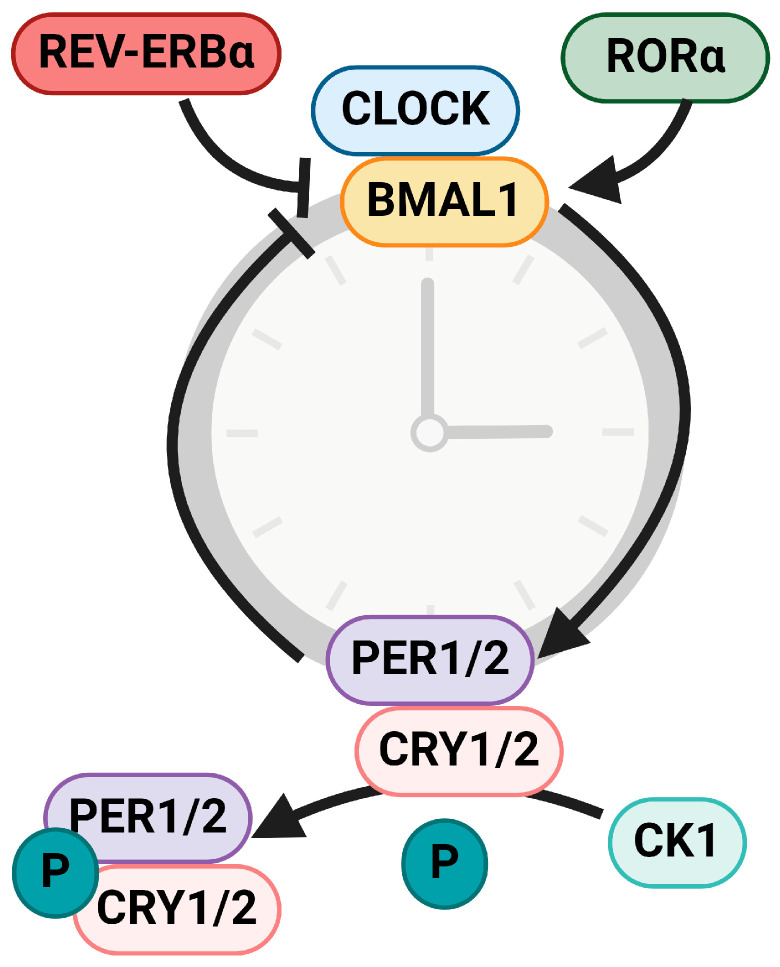
Mammalian circadian rhythms. Circadian rhythms are driven by a transcriptional-translational feedback loop. The core clock proteins CLOCK and BMAL1 heterodimerize and drive gene transcription of *PER1/2* and *CRY1/2*. PER and CRY proteins feedback inhibit CLOCK/BMAL1 interactions, thus inhibiting *PER/CRY* translation. PER and CRY are phosphorylated by CK1, which leads to their ubiquitin-mediated degradation as well as completion of a 24 h cycle. BMAL1 is also regulated by accessory clock proteins; REV-ERBα inhibits *BMAL1* while RORα induces *BMAL1*. Arrowhead indicates pathway activation, flathead indicates inhibition.

**Figure 2 cancers-16-02925-f002:**
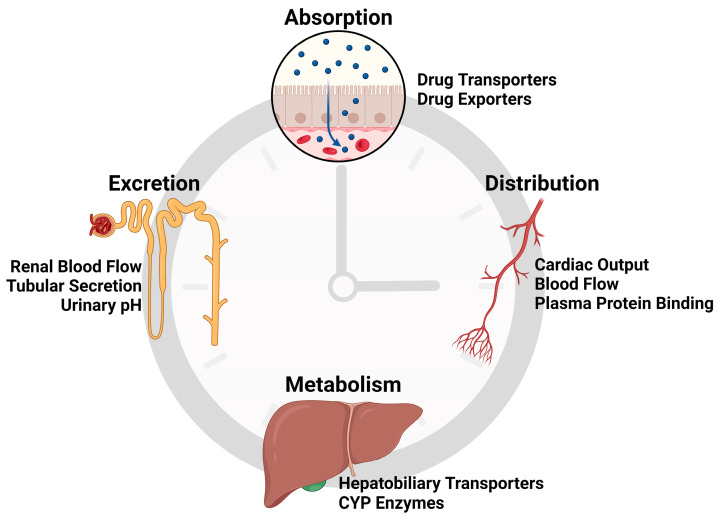
Circadian regulation of drug metabolism. The circadian clock directly or indirectly regulates all aspects of drug absorption, distribution, metabolism, and excretion. Absorptive drug transporters and excretory drug exporters in the intestine and other tissues are regulated by circadian rhythms, and the distribution of drugs is dependent upon rhythmic changes in cardiovascular function and plasma proteins. Drugs metabolized in the liver are subject to circadian regulation of cytochrome P450 enzymes (CYP enzymes) and the hepatobiliary transporters responsible for drug uptake into hepatocytes and secretion into bile. Finally, elimination and excretion of drugs are rhythmically controlled via changes in renal blood flow, tubular secretion rates, and urinary pH.

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
