# Peer review of "Chronobiology of Cancers in the Liver and Gut"

_cancers, 2024, doi:10.3390/cancers16172925_

Round 1

Reviewer 1 Report

Comments and Suggestions for Authors

This is a review article concerning Chronobiology of Cancers in the Liver and Gut

The topic is promising

The article is well written. It is necessary to highlight the usefulness of the clinical application (in a specific section)

Moreover, the abstract must be improved (too generic)

Please consider all the other relevant topic to be compared

For example:

- Circadian Mechanisms in Medicine. N Engl J Med. 2021 Feb 11;384(6):550-561. doi: 10.1056/NEJMra1802337. PMID: 33567194; PMCID: PMC8108270.

Therapeutic Targets and Tumor Microenvironment in Colorectal Cancer. J Clin Med. 2021 May 25;10(11):2295. doi: 10.3390/jcm10112295

Reviewer 2 Report

Comments and Suggestions for Authors

The author reviews comprehensive points where dysregulation of circadian clocks can contribute to inflammation and disease progression, including cancer. It is a well-described review that can serve as a useful guide for readers at the interface between circadian biology and medicine.

I believe the manuscript can be accepted only after adjusting the transcript notations. Circadian biology relies on the transcriptional-translational feedback loop (TTFL), so many components of circadian gene expression are denoted by transcripts (mRNAs). The convention in the field is to italicize the names of transcripts. For example, Per1/2 are denoted Per1/2 (in italics). These errors can be found in lines 132–133 and 175, for example. On the other hand, protein names are written in all capitals and are not italicized. The Bmal1:Clock heterodimer in line 141 should be denoted as the BMAL1:CLOCK protein complex. Additionally, in the manuscript, higher taxonomical ranks such as Clostridiales were not italicized, which is correctly in line with the convention.

After this very minor revision, I believe the manuscript will be ready for acceptance.

Reviewer 3 Report

Comments and Suggestions for Authors

This review article describes the role of circadian rhythms in the pathogenesis and therapy of liver and intestinal cancers.

The article is interesting and well written. I would suggest that the author discuss in more detail the effect of circadian rhythms on bioavailability and pharmacokinetic parameters of drugs used in liver and intestinal cancer therapy.

Comments on the Quality of English Language

Minor editing of English language required.
